# Utility of an Archival Dried Blood Spot (DBS) Collection from HIV-Infected Individuals with and without Cancer in a Resource-Limited Setting

**DOI:** 10.3390/ijms251910235

**Published:** 2024-09-24

**Authors:** Rongzhen Zhang, Paige M. Bracci, Alan Leong, Cassandra Rapp, Michael S. McGrath

**Affiliations:** 1Department of Medicine, University of California San Francisco, San Francisco, CA 94110, USA; 2The AIDS and Cancer Specimen Resource (ACSR), San Francisco, CA 94110, USA; 3Department of Epidemiology and Biostatistics, University of California San Francisco, San Francisco, CA 94110, USA

**Keywords:** archival dried blood spot (DBS), HIV, Kaposi sarcoma-associated herpesvirus (KSHV), Kaposi sarcoma (KS), DNA, inflammatory cytokines, AIDS and Cancer Specimen Resource (ACSR)

## Abstract

The frequency of virus-associated cancers is growing worldwide, especially in resource-limited settings. One of the biggest challenges in cancer research among people living with HIV (PLWH) has been understanding how infection with both HIV and Kaposi sarcoma-associated herpesvirus (KSHV) promotes the pathogenesis of Kaposi sarcoma (KS), the most common cancer among PLWH worldwide and a significant public health problem in regions with high prevalence of HIV such as Sub-Saharan Africa (SSA). The AIDS and Cancer Specimen Resource (ACSR) provides samples for research, including dried blood spots (DBS) that were collected from large clinical epidemiology studies of KSHV and KS in PLWH conducted more than a decade ago in SSA. Here, we validated the quality of DNA derived from DBS samples from SSA studies and provided evidence of quantitative recovery of inflammatory cytokines using these DBS samples through comparison with paired frozen plasma. Significant differences in DNA, protein yields, and inflammatory biomarker levels were also observed between PLWH with/without KS. Establishing the fitness of DBS samples for studies of KS pathogenesis extends the number of projects that can be supported by these ACSR special collections and provides evidence that DBS collection for future KS research is a practical option in resource-limited settings.

## 1. Introduction

Biobanks of human biological samples constitute resources with significant research potential. Obtaining adequate samples to assess feasibility/new concepts as well as sufficient numbers of well-annotated samples needed to test advanced hypotheses and for genomic studies can be challenging. Dried blood spots (DBS) are an excellent and relatively inexpensive source of human biological samples that have been used for serology, drug monitoring, and molecular studies suitable for testing antibodies, antigens, or nucleic acids using most laboratory methods [1,2,3,4,5,6]. DBS are easy to collect and transport compared to the application of conventional protocols for venous blood, especially in resource-limited settings. The collection of DBS has been increasing in clinical practice and for scientific research as a result of new techniques and affordable technologies that have made DBS a feasible option for numerous molecular and genetic studies [7,8,9].

The AIDS and Cancer Specimen Resource (ACSR) was established as a repository of human biological samples collected from people living with HIV (PLWH) by the National Cancer Institute (NCI) in 1994 to encourage and facilitate research on HIV and its associated diseases and conditions [10]. It is the leading resource for well-annotated tissue and biological human samples and its mission is to support investigators conducting HIV-related research, including cancer, virology, immunology, pathology, epidemiology, tumor biology, and assay development.

Infection with HIV has become a chronic condition that can be medically managed for PLWH, yet in lower-resource settings with a high burden of HIV, such as in Sub-Saharan Africa (SSA), morbidity and mortality of PLWH due to HIV is a public health challenge. Further, despite the introduction of antiretroviral therapies (ART), which have greatly diminished the incidence and prevalence of diseases and conditions related to infection with HIV, Kaposi sarcoma (KS) remains one of the most commonly diagnosed cancers in PLWH, especially in SSA where Kaposi sarcoma-related herpesvirus (KSHV), the virus that causes KS, is endemic. The ACSR is custodian of a unique inventory of samples and data donated from closed longitudinal clinical epidemiology studies of KS and KSHV conducted in Uganda. The Antiretrovirals in Kaposi Sarcoma (ARKS) study recruited treatment-naïve PLWH in Uganda with a confirmed diagnosis of KS, who were then randomly assigned to initiate two different antiretroviral therapies, whereas the Uganda AIDS Rural Treatment Outcomes (UARTO) study was a natural history study examining the effects of ART in PLWH without KS. Both studies have been described previously [11]. Briefly, blood (including DBS), saliva, and tissue (ARKS only) as well as clinical and epidemiological data were collected from participants at baseline prior to ART and for up to 5 years post ART to assess their response to ART over time, including the evolution of KS and HIV infection. Samples, including DBS, have been stored at −80 °C for more than a decade although the storage conditions for DBS samples varied over time. To increase the number of research projects that can be supported by these specimen collections, “sample sparing” activities have been implemented, e.g., the creation of tissue microarrays and extraction of nucleic acids from fluids and tissue, as well as the assessment of sample fitness for use using new techniques and technologies. The collection and use of DBS samples as a potential alternative to venous blood and blood derivatives within a biobanking resource is understudied and holds promise for use in various analyses such as molecular-based/genomic, inflammatory biomarker, and metabolic profile studies.

The purpose of the current pilot study is to determine the quality of archival DBS samples that were stored variably at room temperature and/or −80 °C compared with plasma collected from the same patients at the same time and stored at −80 °C. To accomplish this, we extracted and tested the quality of DNA and specific proteins related to the inflammatory process from the archival ARKS and UARTO DBS samples.

## 2. Results

### 2.1. DNA Yield from DBS

A total of 24 archival DBS samples collected in 2011, 12 ARKS and 12 UARTO were evaluated. DNA yields varied between DBS samples and study collections. The overall DNA yield (mean ± SD) of the DBS samples was 138.0 ± 47.0 ng (57.0–217.5 ng). DNA yields from ARKS DBS were significantly higher than those from UARTO DBS (unpaired *t*-test, *p* = 0.003) (Figure 1).

### 2.2. DNA Quality in the DBS

The integrity of DNA was determined using Agilent 2100 Bioanalyzer. As shown in Figure 2, the DNA samples from DBS were running into the upper marker (10,380 bp) in Bioanalyzer electrophoresis, which will give the incorrect sizing and quantification. However, the results from Bioanalyzer show that the majority of genomic DNA (gDNA) produced from DBS had relatively high molecular weight.

### 2.3. Protein Concentration from DBS

Protein samples were extracted from the same set of 24 archival DBS samples, 12 ARKS and 12 UARTO. Protein yields also varied between DBS samples and collections. The overall concentration (mean ± SD) from 1/3 of a DBS was 5793 ± 1015 μg/mL (4107–7988 μg/mL). Similar to DNA yield, protein concentrations from ARKS DBS were significantly higher than those from UARTO DBS (unpaired *t*-test, *p* = 0.01) (Figure 3).

### 2.4. Relationship between DNA Yield and Protein Concentration

A direct correlation between DNA yield and protein concentration from the same DBS samples was observed from 24 archival DBS samples (correlation analysis, r = 0.492, *p* = 0.01, *n* = 24), as shown in Figure 4.

### 2.5. Biomarker Assays in DBS Extraction Supernatants and Paired Plasma Samples

A total of 16 biomarkers were assessed in DBS and paired plasma. The biomarkers evaluated included (1) pro-inflammatory cytokines: IL-1β, IL-6, TNF-α, IL-2, IL-12p70, IL-17A, & IFN-g; (2) anti-inflammatory cytokines: IL-4, IL-10 & IL-13; (3) chemokines for peripheral immune cell recruitment: monokine induced by gamma interferon (MIG), RANTES, MCP-1 & IL-8; (4) CRP and IL-2Ra. Biomarker levels for DBS extraction supernatants from the same set of 12 ARKS and 12 UARTO DBS and paired plasma samples were evaluated at the same time.

#### 2.5.1. Relationship of Inflammatory Biomarker Levels between Archival DBS Extraction Supernatants and Paired Plasma Samples

There was a positive correlation between DBS extraction supernatant and paired plasma samples for log-transformed levels of six biomarkers (Figure 5) (correlation analysis, *n* = 24); MIG (r = 0.417, *p* = 0.04), CRP (r = 0.910, *p* < 0.0001), RANTES (r = 0.552, *p* = 0.005), INF-g (r = 0.524, *p* = 0.009), IL-8 (r = 0.450, *p* = 0.03), IL-17 (r = 0.820, *p* < 0.0001).

#### 2.5.2. High Levels of Biomarkers Observed in DBS Extraction Supernatants

With more than a 10-fold dilution, the log-transformed levels of four biomarkers in DBS extraction supernatants were significantly higher than in paired plasma samples (Figure 6) (paired *t*-test, *n* = 24); RANTES (*p* < 0.0001), IL-1b (*p* < 0.0001), IL-8 (*p* < 0.0001), and IL-2Ra (*p* = 0.01).

### 2.6. Differences in Levels of Inflammatory Biomarkers between UARTO and ARKS DBS Extraction Supernatants and Paired Plasma Samples

#### 2.6.1. Differences between UARTO and ARKS Inflammatory Biomarker Levels in DBS Extraction Supernatants

As shown in Figure 7, the log-transformed levels of four inflammatory biomarkers from DBS extracted supernatants were significantly higher in samples from ARKS as compared to UARTO participants (Figure 7) (unpaired *t*-test); RANTES (*p* = 0.03), MIG (*p* = 0.03), INF-g (*p* = 0.05), and IL-1b (*p* = 0.03).

#### 2.6.2. Inflammatory Biomarker Differences between UARTO and ARKS Plasma Samples

The log-transformed levels of six inflammation-related biomarkers were significantly elevated in ARKS compared to UARTO plasma samples (Figure 8); MCP-1 (*p* = 0.02), MIG (*p* = 0.03), TNF-a (*p* < 0.001), IL-10 (*p* = 0.01), IL-2 (*p* = 0.002), and IL-8 (*p* = 0.007) (unpaired *t*-test). Note that MIG was also higher in ARKS DBS extraction supernatants than in UARTO.

## 3. Discussion

Our findings show that good-quality high molecular weight DNA can be extracted from more than decade-old African DBS samples stored under variable conditions. Approximately 57.0 ng–217.5 ng of high molecular weight DNA was obtained from one-third of a dried blood spot (~17 μL of EDTA blood), which might be sufficient for PCR-based applications (>50 ng of DNA is standard) [12,13,14] and is an adequate quantity for other downstream applications, such as host and genomic virus sequencing and SNP genotyping. Further, the good yields of protein extracted from these archival DBS samples provide support for DBS as a potentially valuable medium for the evaluation of inflammatory biomarkers. However, given that the variability in biomarker results and the presence of other components in whole blood might interfere with the quantification of some cytokines/chemokines [15,16,17,18], the broad use of DBS in protein studies requires the validation of each marker.

Significantly higher DNA and protein yields were found in archival DBS samples in the PLWH with confirmed KS (ARKS) as compared to those without KS (UARTO). DBS DNA yield showed a positive correlation with DBS protein concentration. The higher white blood cells fighting off infection or inflammation could be the reason for the high yields of DNA and proteins in the overall cohort with confirmed KS [19,20].

The levels of inflammatory biomarkers were significantly higher in the ARKS KS study collection as compared to the individuals in the HIV-positive and non-KS UARTO cohort. Given the fact that the sample sizes were limited in this pilot study and the correlations were observed for many biomarkers between DBS extraction supernatants and paired plasma samples, the archival African DBS samples may still be used as a potential resource for studies of inflammation and the effect of antiretroviral treatment in HIV-infected adults with confirmed KS in Africa (ARKS).

The levels of some cytokines/chemokines were higher in DBS extraction supernatants than paired blood plasma even with more than 10-fold dilutions. Those biomarkers could be endogenous proteins from other components of whole blood in DBS. Besides plasma, the other three components in whole blood, platelets, white blood cells, and red blood cells, also contain and release a significant number of cytokines and chemokines [21,22,23]. Differently from plasma, DBS samples were collected from EDTA whole blood. The cytokines/chemokines may be released from non-plasma components during cell lysis in DBS protein extraction, resulting in a significant increase in some cytokines/chemokines. If this is proven to be true, DBS samples may be a better resource for the evaluation of those biomarkers than plasma samples. For example, given the significant signals derived from the cellular components in DBS, different from factor levels in plasma, an unexpectedly high cellular signal in a UARTO specimen might be an indication of a KS-like process unappreciated at the time of blood draw. Further analyses are warranted to more thoroughly address this question.

In African longitudinal epidemiologic studies, serial DBS samples are available for the majority of participants in both the ARKS and UARTO cohorts. The archival DBS samples might be good resources for the research of proteins and inflammatory biomarkers [24,25]. They also might be appropriate for studies of small molecule biomarkers, such as metabolic profile studies where only a small amount of material is needed and the measurement of small molecules has been shown to be stable even at room temperature [26,27,28,29]. Overall, the archival African DBS samples might be a promising resource for various studies of disease mechanisms as well as the assessment of HIV drugs on HIV infection and KS in Africa and other studies in resource-limited settings, especially in those where the availability of liquid samples is limited.

Guthrie introduced DBS samples for wider use more than 60 years ago for the neonatal screening of inborn errors of metabolism (initially phenylketonuria [30]). Since then, DBS samples have been collected and stored in biobanks to conduct field epidemiological studies worldwide. DBS samples have been used extensively in both human healthcare and research that ranges from newborn screening to epidemiological studies in low-resource settings (such as the studies that supported these analyses) [31,32,33,34,35,36,37]. Compared to conventional blood samples, DBS collection can be a practical, affordable and easier-to-implement method for blood collection in epidemiological and clinical research studies. With the advent of improved assays that require smaller quantities/volumes of material for analysis, DBS is now a feasible alternative in a wide variety of research studies as well as for biospecimen archiving.

## 4. Materials and Methods

### 4.1. Collection and Storage of Biobanking Samples

Blood collected in EDTA tubes, was spotted on 903 Whatman protein saver cards (Sigma-Aldrich, Inc., St. Louis, MO, USA). A total of 50 μL of EDTA blood for each spot and 150 μL on 3 spots were collected on one protein saver card for each individual sample collection. A total of more than 22,000 DBS aliquots were collected longitudinally in these studies, 12% of which were collected from 2007 to 2012 as part of the ARKS study, and 88% were collected from 2010 to 2014 as part of the UARTO study.

The DBS cards were stored at room temperature in plastic Ziploc bags with desiccant from anywhere between 3 months and 1.5 years, when they were then shipped to the ACSR University of California San Francisco (UCSF) regional biospecimen repository in San Francisco, California. Upon receipt, the DBS samples were placed in −80 °C freezers for long-term storage.

In this pilot study, a total of 24 archival DBS samples were evaluated that had been collected in 2011 from 12 ARKS and 12 UARTO participants at time points post-ART initiation. The paired plasma samples used as biomarker references for this study were processed in Uganda from venous blood collected in EDTA tubes that were aliquoted and stored in −80 °C freezers immediately after processing. Basic demographic and clinical data for these 24 participants are shown in Table 1. All 24 participants were on stable ART.

### 4.2. Genomic DNA (gDNA) Extraction from DBS

A section that was 1/3 of a DBS was cut from each spot/aliquot using a sterile single-edge razor blade in the biosafety cabinet. The QIAGEN QIAamp DNA Investigator kit (QIAGEN, Valencia, CA, USA) was used for DNA extractions according to the manufacturer’s instructions with a slight modification as follows: 1/3 of a DBS was first mixed with 280 μL Buffer ATL and 20 μL Protein K and shaken at 900 rpm and 56 °C for 1 h in a dry bath shaker. The sample was centrifuged briefly after incubation and transferred to a new microcentrifuge tube, to which 300 μL of the Buffer AL and carrier RNA was added and then incubated at 70 °C with shaking at 900 rpm for 10 min. A total of 150 μL of ethanol was added after briefly centrifuging and mixing completely, and the mixture was transferred to a QIAamp MinElute column and centrifuged at 8000 rpm for 1 min. Then, a two-step wash/centrifuge was performed using AW1 solution in the first step and AW2 solution in the second step. Ethanol residue was further removed by centrifugation at 14,000 rpm for 3 min followed by incubation at room temperature for 10 min. For the last step of DNA collection, 50 µL of buffer ATE solution was applied and incubated for 5 min before centrifuging at 14,000 rpm for 1 min; this elution step was repeated for the collected solution and the first eluate was collected. A total of 30 µL of buffer ATE solution was applied using the same procedure and the second eluate was collected. The first eluate was combined with the second eluate resulting in a total volume of approximately 75 μL of DNA solution.

### 4.3. Assessment of gDNA from DBS

#### 4.3.1. Qubit Fluorometer 3.0

Qubit fluorometer 3.0 (Invitrogen, Waltham, MA, USA) analysis was performed to measure the genomic DNA concentration. A Qubit 1x dsDNA HS (high sensitivity, 0.2 to 100 ng) Assay Kit (Invitrogen, Waltham, MA, USA) was used with a Qubit 3.0 fluorometer according to the manufacturer’s instructions; a sample volume of 1 μL was added to 199 μL of a Qubit working solution before measurement.

#### 4.3.2. Agilent Bioanalyzer 2100

As DNA yields varied, quantification and sizing of gDNA was performed by capillary electrophoresis (CE) using an Agilent 2100 Bioanalyzer (Agilent Technologies Inc., Santa Clara, CA, USA) equipped with Expert 2100 software (Agilent, v. B.02.11.SI824), in combination with Agilent High Sensitivity (HS) DNA kit, which can resolve the size of high molecular weight DNA up to 10,380 bp. The assay was performed according to the instructions provided by the manufacturer. The parallel extraction of empty/blank spots on the same set of DBS protein saver cards was run in the Bioanalyzer at the same time as an extraction control.

### 4.4. Protein Extraction from Archival DBS and Concentration Measurement

#### 4.4.1. Protein Extraction Buffer

Protein extraction buffer was prepared immediately before use. Briefly, 10 mL of D-PBS (Dulbecco’s Phosphate-Buffered Saline, UCSF Cell Culture Facility, San Francisco, CA, USA) and 50 μL of Tween 20 (0.5%) (Sigma-Aldrich, Burlington, MA, USA) and 1 tablet of cOmplete, Mini EDTA-free Protease Inhibitor Cocktail (Roche, Indianapolis, IN, USA) were combined and mixed well.

#### 4.4.2. Protein Extraction

A total of 300 μL of protein extraction buffer was added to each DBS sample (1/3 of a DBS) ensuring that all DBS pieces were submerged in the extraction buffer. The mixture was shaken at 600 rpm for 1 h at room temperature in a dry bath shaker followed by centrifugation at 14,000 rpm for 30 s, after which the supernatant was collected. The final supernatant volume was approximately 230 μL.

#### 4.4.3. Protein Quantification

A Pierce BCA Protein Assay Kit (Thermo Scientific, Waltham, MA, USA) was used for protein quantification according to the manufacturer’s instructions. Samples were read at a wavelength of 562 nm with a Biotek Cytation3 microplate reader (BioTek, Winooski, VT, USA).

### 4.5. Factors Evaluated in DBS Extraction Supernatants and Paired Plasma Samples

Biomarker levels for DBS extraction supernatants and paired plasma samples were evaluated at the same time by the commercial company AssayGate, Inc. (Ijamsville, MD, USA). Empty/blank spots from the same sample set of DBS protein saver cards were used for the calibrator extraction control. The 16 inflammation-associated biomarkers evaluated included: IL-1β, IL-6, TNF-α, IL-2, IL-12p70, IL-17A, IFN-g, IL-4, IL-10, IL-13, Monokine induced by gamma interferon (MIG), RANTES, MCP-1, IL-8, CRP and IL-2Ra.

### 4.6. Statistical Analysis

Data were analyzed using JMP Pro 17 (SAS Institute, Cary, NC, USA). The unpaired *t*-test was used to compare DNA and protein yields between two different study groups. A paired *t*-test was used to compare biomarker levels in DBS extraction supernatants and paired plasma samples. Correlation analysis was performed to analyze the relationship between DNA and protein yields and the relationship of biomarker levels between DBS extraction supernatants and paired plasma samples. Data that did not meet normality assumptions were log-transformed prior to analysis. For all analyses, a two-sided *p*-value < 0.05 was considered statistically significant.

## 5. Conclusions

The ACSR provides high-quality specimens for research that were collected from PLWH who were part of large clinical epidemiology studies of KSHV and KS conducted in Sub-Saharan Africa, including the Antiretrovirals in KS (ARKS) and Uganda AIDS Rural Treatment Outcomes (UARTO) studies. KS remains a leading cause of death in PLWH in Africa, so having both the KS patient samples from ARKS and control samples from UARTO for pathogenesis-related research is invaluable. The validation of this DBS longitudinal collection from more than a decade ago expands the availability of both DNA and protein for biomarker studies relevant to this field. Further research on the archival DBS samples is required to increase our knowledge and to identify the full potential of these archival samples in KS and KSHV research in PLWH.

## Figures and Tables

**Figure 1 ijms-25-10235-f001:**
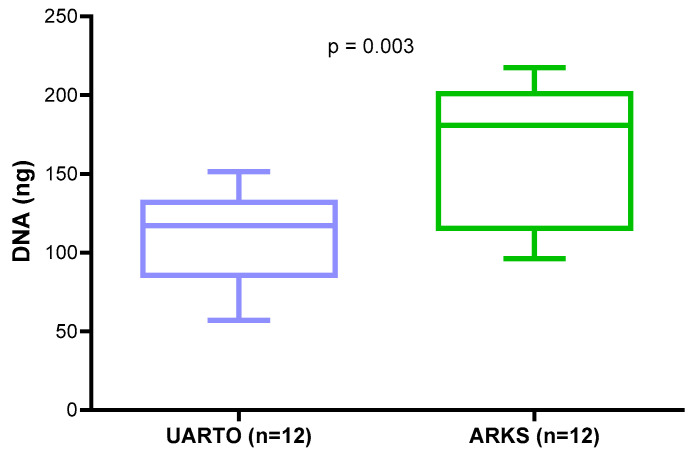
DNA yield comparison between UARTO and ARKS cohorts. Box-and-whisker plots depict the DNA yields from DBS for UARTO and ARKS cohorts. Significantly higher DNA yields from people living with HIV (PLWH) with confirmed Kaposi sarcoma (KS) (ARKS, *n* = 12, in green) as compared to non-KS PLWH cohort (UARTO, *n* = 12, in blue) (unpaired *t*-test, *p* = 0.003).

**Figure 2 ijms-25-10235-f002:**
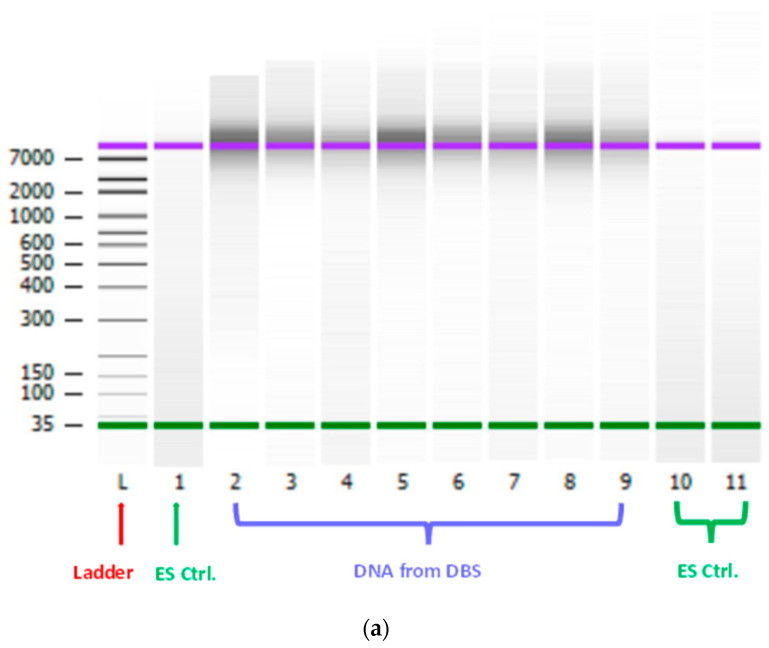
(**a**) Gel image example of DNA from DBS from the Agilent 2100 Bioanalyzer. Lane L, ladder; Lanes 1, 10 and 11, DNA extraction controls isolated from empty/blank spots (ES Ctrl.); Line 2–9, DNA samples isolated from DBS. (**b**) Histogram example of empty/blank spot control samples (Line 1, ES Ctrl.). (**c**) Histogram example of DNA samples isolated from DBS (Line 2, DNA from DBS).

**Figure 3 ijms-25-10235-f003:**
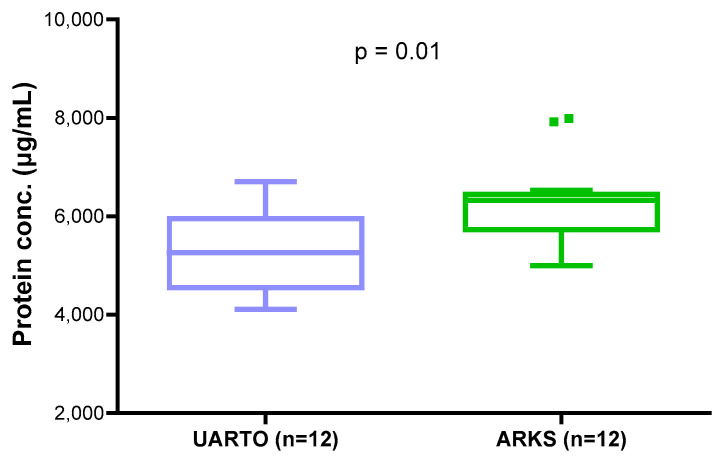
Protein yield comparison between UARTO and ARKS cohorts. Box-and-whisker plots depict the protein concentrations from DBS for UARTO and ARKS cohorts. Significant higher protein concentrations were observed in people living with HIV (PLWH) with confirmed Kaposi sarcoma (KS) (ARKS, *n* = 12, in green) as compared to the non-KS PLWH cohort (UARTO, *n* = 12, in blue) (unpaired *t*-test, *p* = 0.01).

**Figure 4 ijms-25-10235-f004:**
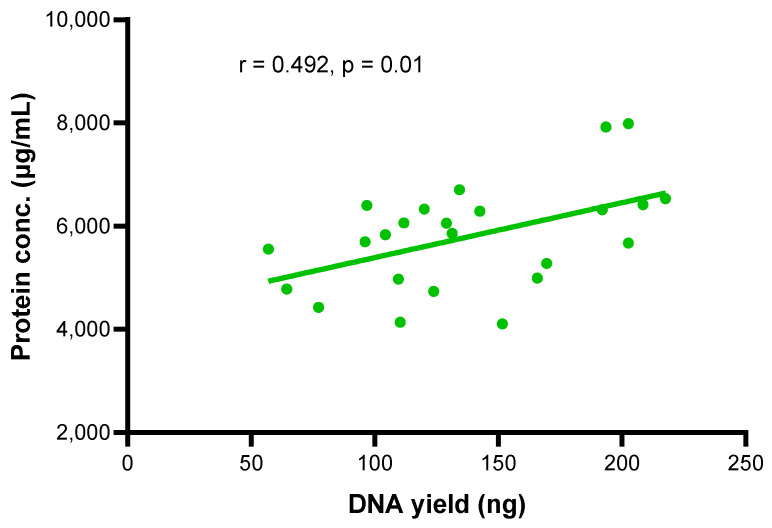
The direct relationship between DNA yield and protein concentration from the same DBS samples (correlation analysis, r = 0.492, *p* = 0.01, *n* = 24).

**Figure 5 ijms-25-10235-f005:**
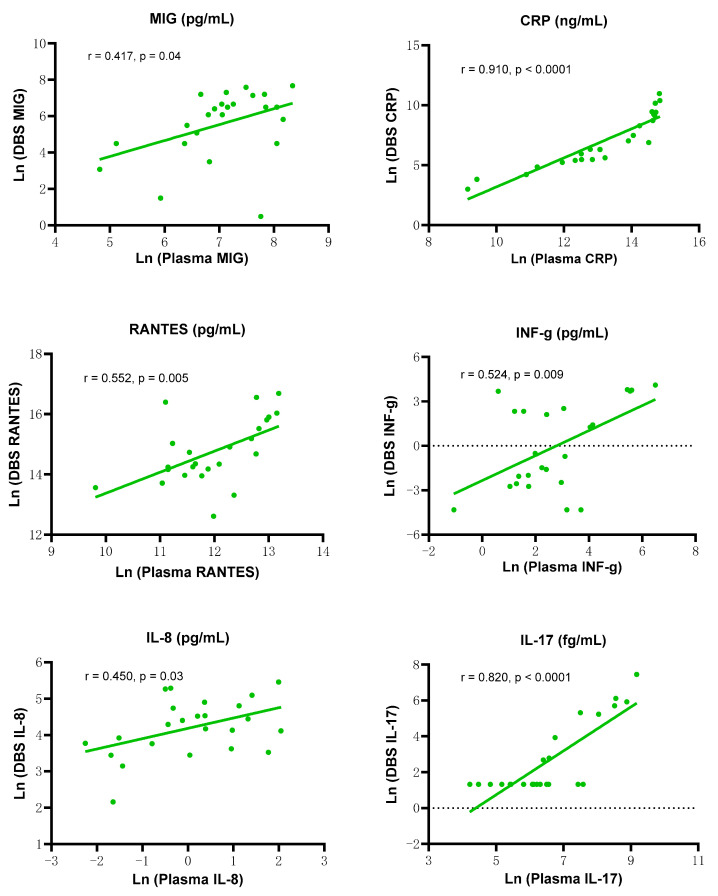
Examples of relationships between DBS extraction and paired plasma biomarker levels. A positive correlation was observed between log-transformed biomarker levels from DBS extraction supernatants and paired plasma samples (correlation analysis, *n* = 24); MIG (r = 0.417, *p* = 0.04), CRP (r = 0.910, *p* < 0.0001), RANTES (r = 0.552, *p* = 0.005), INF-g (r = 0.524, *p* = 0.009), IL-8 (r = 0.450, *p* = 0.03), IL-17 (r = 0.820, *p* < 0.0001).

**Figure 6 ijms-25-10235-f006:**
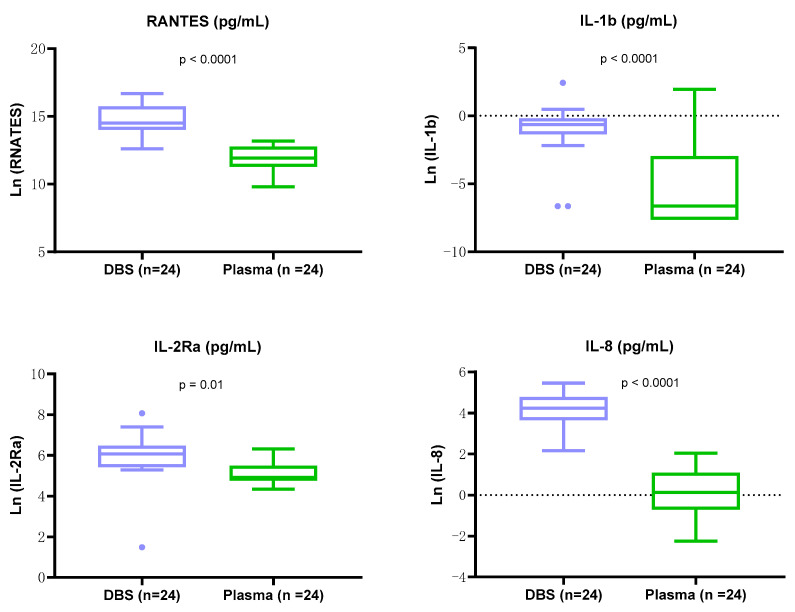
High levels of biomarkers observed in DBS extraction supernatants. Box-and-whisker plots depict log-transformed biomarker levels for DBS extraction supernatants (n = 24, in blue) and paired plasma samples (*n* = 24, in green). There were significantly higher levels of four biomarkers in DBS extraction supernatants (paired *t*-test), RANTES (*p* < 0.0001), IL-1b (*p* < 0.0001), IL-2Ra (*p* = 0.01), and IL-8 (*p* < 0.0001).

**Figure 7 ijms-25-10235-f007:**
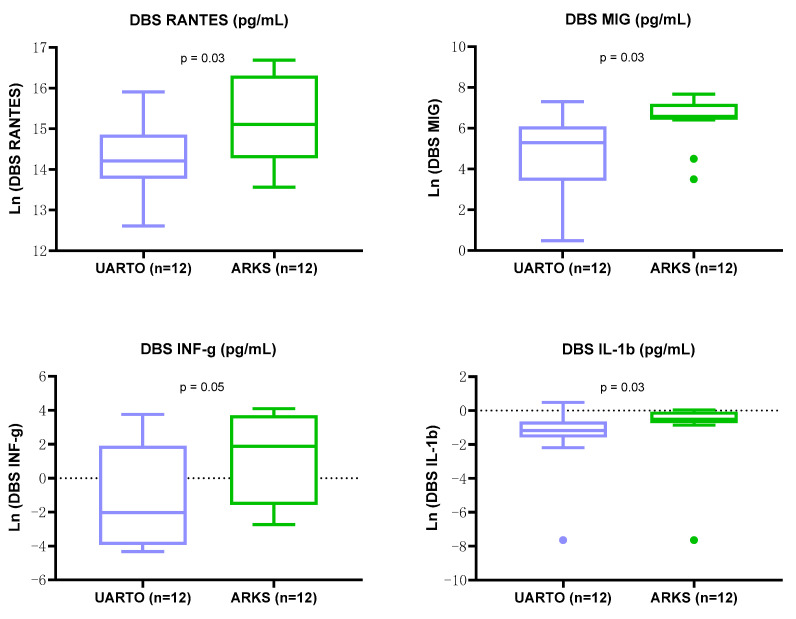
Log-transformed levels of inflammation-associated biomarkers from DBS extraction supernatants were increased in ARKS compared to UARTO participants. Box-and-whisker plots depict the log-transformed biomarker levels of DBS extraction supernatants for UARTO (*n* = 12, in blue) and ARKS (*n* = 12, in green). Levels of RANTES (*p* = 0.03), MIG (*p* = 0.03), INF-g (*p* = 0.05), and IL-1b (*p* = 0.03) were significantly higher in DBS extraction supernatants from the ARKS than from UARTO (unpaired *t*-test).

**Figure 8 ijms-25-10235-f008:**
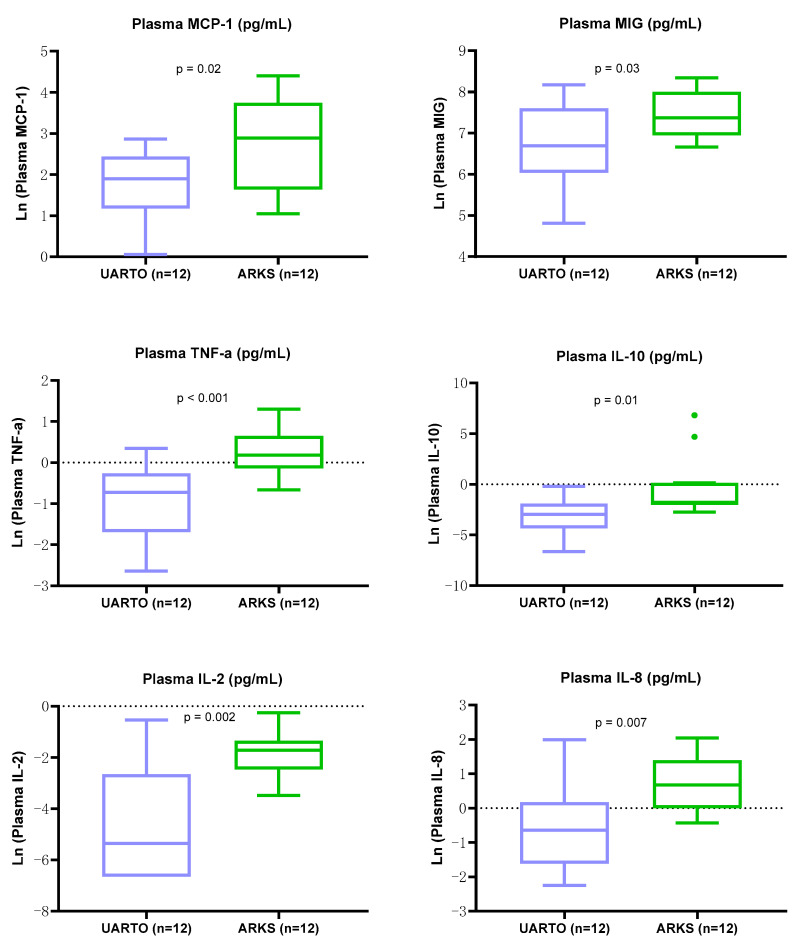
Log-transformed levels of KS/inflammation-associated biomarkers were higher in ARKS than in UARTO plasma samples. Box-and-whisker plots depict log-transformed biomarker levels in UARTO (*n* = 12, in blue) and ARKS (*n* = 12, in green) plasma samples. The six log-transformed biomarkers with statistically significantly higher levels in ARKS than in UARTO are MCP-1 (*p* = 0.02), MIG (*p* = 0.03), TNF-a (*p* < 0.001), IL-10 (*p* = 0.01), IL-2 (*p* = 0.002), IL-8 (*p* = 0.007) (unpaired *t*-test).

**Table 1 ijms-25-10235-t001:** Basic demographic and clinical characteristics at the time of sample collection for 12 antiretrovirals in Kaposi sarcoma (ARKS) and 12 Uganda AIDS rural treatment outcomes (UARTO) study participants included in these analyses.

Study Cohort	N	Sex	Mean Age in Years ± SD	Race	HIV Status	KS Status
UARTO	12	8 Female/4 Male	43.3 ± 7.1	Black African	HIV positive	Non-KS
ARKS	12	4 Female/8 Male	36.8 ± 9.2	Black African	HIV positive	Confirmed KS

## Data Availability

The data are available through ACSR upon request.

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
