# Peer review of "Utility of an Archival Dried Blood Spot (DBS) Collection from HIV-Infected Individuals with and without Cancer in a Resource-Limited Setting"

_ijms, 2024, doi:10.3390/ijms251910235_

Round 1

Reviewer 1 Report

Comments and Suggestions for Authors

I would move the materials and methods section after the introduction rather than at the end of the work because it is always better to have the conclusions at the end. I would also better emphasize the discourse relating to the differences in the content of the ARKS and UARTO data, adding hypotheses with respect to the possibility of using the data from the samples to hypothesize a subclinical Kaposi, for example.

Author Response

I would move the materials and methods section after the introduction rather than at the end of the work because it is always better to have the conclusions at the end. I would also better emphasize the discourse relating to the differences in the content of the ARKS and UARTO data, adding hypotheses with respect to the possibility of using the data from the samples to hypothesize a subclinical Kaposi, for example.

Response to Reviewer 1 comments:

  1. In accordance with the journal's guidelines, which specify that sections should be ordered as follows: 1. Introduction; 2. Results; 3. Discussion; 4. Materials and Methods; 5. Conclusions. We have modified and moved the last paragraph in the “Discussion” to the end of the manuscript as “Conclusions” in the revised version.
  2. To emphasize the discourse relating to the differences in the content of the ARKS and UARTO data as the reviewer suggested, the following sentence of hypotheses has been added in the fourth paragraph in the “Discussion” section. “For example, given the significant signals derived from the cellular components in DBS, different from factor levels in plasma, an unexpectedly high cellular signal in a UARTO specimen might be an indication of a KS like process unappreciated at the time of blood draw.”

Reviewer 2 Report

Comments and Suggestions for Authors

This is an interesting proposal with health implications.

Minor comments and suggestions:
- Abstract section: 1) Please change "person living with HIV" to "people living with HIV"; 2) define KSHV
https://www.unaids.org/sites/default/files/media_asset/2024-terminology-guidelines_en.pdf
- Introduction section: It is adequate. The authors have done an appropriate approach to the problem.
- Results section: It is adequate. The images provide relevant information.
- Discussion section: Although it is appropriate, it would be advisable to include some references to the potential limitations as well as the advantages of this technique.
- M&M section: It is adequate. The authors details in an adequate way the technology employed.
- Reference section: It is adequate.

Author Response

Minor comments and suggestions:
- Abstract section: 1) Please change "person living with HIV" to "people living with HIV"; 2) define KSHV
https://www.unaids.org/sites/default/files/media_asset/2024-terminology-guidelines_en.pdf

Thanks to the reviewer for the information. We have defined KSHV in the abstract and made changes to "people living with HIV” in (1) Abstract section, (2) Instruction section: the first description in the second paragraph, and (3) Figure legends for Figures 1 & 3.

- Introduction section: It is adequate. The authors have done an appropriate approach to the problem.
- Results section: It is adequate. The images provide relevant information.
- Discussion section: Although it is appropriate, it would be advisable to include some references to the potential limitations as well as the advantages of this technique.

As suggested by the reviewer, we have included more information and a few references to the potential limitations and advantages for the use archival DBS in the first and fourth paragraphs in “Discussion” section in the revised version.

- M&M section: It is adequate. The authors details in an adequate way the technology employed.
- Reference section: It is adequate.

Reviewer 3 Report

Comments and Suggestions for Authors

Congratulations to the authors for the proposal presented. There is no doubt that the use of DBS samples is an excellent strategy for collecting and storing biological specimens and subsequently analyzing various markers, both from the host and from various infectious agents. Below I present a few suggestions/questions for the authors of the manuscript.

1- Which study did the samples used in the correlation between DNA/protein (Figures 4 and 5) come from? (UARTO or ARKS). I was unable to identify this information in the text;

2- In the methodology, the authors mention that the 24 samples used in the study were collected in 2011. Did the authors consider performing a comparison between the oldest samples (e.g., 2007) and the most recent samples (2014)?

3- In addition to analyzing the integrity and yield of DNA and the concentration of proteins in the DBS samples, would it be feasible and relevant to perform the same analysis for RNA? I believe it would be interesting for analyzing gene expression of various markers, as well as detecting other RNA viruses;

4- How long had the DBS samples used to compare the levels of inflammatory and anti-inflammatory markers been stored? The authors report that they had DBS samples stored for a short and long time. Which ones were used in these comparisons? Perhaps this information should be made clearer when presenting the results.

5- What could justify this increased level in the DBS samples compared to the plasma samples? The authors briefly describe it in the discussion, but I think this is one of the most intriguing results of the study. It would be important to further discuss this.

6- Likewise, finding significant differences in the levels of biomarkers between the group with KS and the group without KS was perhaps already expected, since KSHV infection and consequently the development of sarcoma will intensify the inflammatory response. Therefore, I think it would have been interesting to have compared the analysis of all markers in DBS samples at different collection times and storage conditions.

Author Response

1- Which study did the samples used in the correlation between DNA/protein (Figures 4 and 5) come from? (UARTO or ARKS). I was unable to identify this information in the text;

Both UARTO and ARKS were used in the current study, as described in the “Materials and Methods” section. Since the “Materials and “Methods” section was at the end of the manuscript in accordance with the journal's guidelines, we have added the sample information in the “Results” sections 2.1, 2.3, & 2.5 in the revised version, as the reviewer suggested.

2- In the methodology, the authors mention that the 24 samples used in the study were collected in 2011. Did the authors consider performing a comparison between the oldest samples (e.g., 2007) and the most recent samples (2014)?

Definitely. This is a pilot study to determine the quality of archival DBS samples. Further studies will include DBS quality in the different collection time and storage conditions, and KSHV pathogenesis and treatment effect on HIV and KS with this DBS longitudinal collection.

3- In addition to analyzing the integrity and yield of DNA and the concentration of proteins in the DBS samples, would it be feasible and relevant to perform the same analysis for RNA? I believe it would be interesting for analyzing gene expression of various markers, as well as detecting other RNA viruses;

Agree. With very limited samples, we decided to start with DNA and protein analysis because many references have shown the yield and quality of RNA dramatically degraded after long-term storage. It is definitely worth exploring the potential utility of RNA from archival DBS samples, this will be one of our future research projects.

4- How long had the DBS samples used to compare the levels of inflammatory and anti-inflammatory markers been stored? The authors report that they had DBS samples stored for a short and long time. Which ones were used in these comparisons? Perhaps this information should be made clearer when presenting the results.

The DBS samples, both ARKS and UARTO, used in the current study were collected in 2011, as described in the “Materials and Methods”. As the reviewer suggested, the information has been added in the “Results” sections 2.1, 2.3, & 2.5 in the revised version to make it clear.

5- What could justify this increased level in the DBS samples compared to the plasma samples? The authors briefly describe it in the discussion, but I think this is one of the most intriguing results of the study. It would be important to further discuss this.

Thanks for the suggestions. Further discussion and a few references have been added in the fourth paragraph of “Discussion” section in the revised version.

6- Likewise, finding significant differences in the levels of biomarkers between the group with KS and the group without KS was perhaps already expected, since KSHV infection and consequently the development of sarcoma will intensify the inflammatory response. Therefore, I think it would have been interesting to have compared the analysis of all markers in DBS samples at different collection times and storage conditions.

Completely agree! Further studies to answer all these interesting and important questions will be our research projects in the future.